# Flight Capability and the Low Temperature Threshold of a Chinese Field Population of the Fall Armyworm *Spodoptera frugiperda*

**DOI:** 10.3390/insects13050422

**Published:** 2022-04-29

**Authors:** Hui Chen, Yao Wang, Le Huang, Chuan-Feng Xu, Jing-Hui Li, Feng-Ying Wang, Wei Cheng, Bo-Ya Gao, Jason W. Chapman, Gao Hu

**Affiliations:** 1Department of Entomology, Nanjing Agricultural University, Nanjing 210095, China; 2019202044@njau.edu.cn (H.C.); 2018102070@njau.edu.cn (Y.W.); 2019102077@njau.edu.cn (L.H.); 2020202039@njau.edu.cn (C.-F.X.); 2018802171@njau.edu.cn (J.-H.L.); gaoby@njau.edu.cn (B.-Y.G.); j.chapman2@exeter.ac.uk (J.W.C.); 2State Key Laboratory of Biological Interactions and Crop Health, Nanjing Agricultural University, Nanjing 210095, China; 3Institute of Plant Protection, Guangxi Academy of Agricultural Science, Nanning 530007, China; wfy58260396@163.com; 4Shanghai Agricultural Technology Extension and Service Center, Shanghai 201103, China; chengweizb@sina.com; 5Centre for Ecology and Conservation, and Environment and Sustainability Institute, University of Exeter, Penryn, Cornwall TR10 9FE, UK

**Keywords:** fall armyworm, field population, flight ability, low temperature threshold of flight

## Abstract

**Simple Summary:**

The fall armyworm (FAW), an invasive migratory pest from the Americas, has been rapidly spreading through the Old World recently. Although it cannot survive winter periods in temperate and subtropical regions, adults re-invade these areas each spring or summer by virtue of their migratory capability. Therefore, it is important to evaluate the flight capability of FAW moths for monitoring and predicting the range and timing of this migration. In this study, we demonstrate that the potential migration duration (and thus distance) of a field population of FAW from South China is significantly greater than previously estimated. A strong migratory tendency was observed in 58% of individuals, and the longest self-powered flight distance was 116.7 km with a cumulative flight duration of 36.51 h during a 48-h period. Furthermore, our study documented that the low temperature threshold for flight of FAW was 13.1 °C. The results of this paper will be helpful to further understand the migratory rules and flight behavior of FAW, and to provide a theoretical basis for pest forecasting and pest control.

**Abstract:**

The fall armyworm, *Spodoptera frugiperda* (J. E. Smith), is capable of long-distance migration; thus, evaluation of its flight capability is relevant to the design of monitoring and control strategies for this pest. Previous studies have quantified the flight ability of lab-reared populations under controlled conditions, but less is known about the flight capability of natural populations. In addition, the low temperature threshold for flight in natural populations also needs to be determined. In this study, the flight capability of *S. frugiperda* adults emerging from field-collected larvae in South China was measured by a flight mill system. The results show that the flight capability of *S. frugiperda* moths varied greatly between individuals, and that some adults are capable of flying great distances. The longest self-powered flight distance was 116.7 km with a cumulative flight duration of 36.51 h during a 48-h period. Typically, the flight activity of tethered individuals was relatively stable during the first 12 h, indicating that migrating moths can fly through an entire night. Based on the accumulated flight duration in the first 12 h, moths can be clearly divided into two groups (<5 h and ≥5 h flight duration), and 58% of individuals belonged to the latter group with strong migratory tendency. Further, flight activity under low temperature conditions was tested, and the results of a logit generalized linear model indicate that the low temperature flight threshold of *S. frugiperda* is 13.1 °C under declining temperatures. Our results provide a scientific basis for further elucidating the flight biology and migration mechanism of *S. frugiperda*.

## 1. Introduction

The fall armyworm, *Spodoptera frugiperda* (J. E. Smith), an invasive migratory pest from the Americas, is rapidly spreading through many countries in Africa, Asia and Australasia [1,2,3]. This species has a very wide host range (it has been recorded on at least 353 species of plants), but the crop most seriously affected is corn (maize): between 15 and 73% of the yield of corn can be lost due to larval feeding damage [4]. Due to this high level of damage, combined with the strong flight ability of *S. frugiperda*, this species is a great threat to corn production in all regions of the world.

*Spodoptera frugiperda* cannot survive winter periods in temperate and subtropical regions [1,2,5]. However, due to its migratory capability, adults re-invade these areas each spring or summer by migrating from winter-breeding regions [6]. For example, *S. frugiperda* has already formed a seasonal migration pattern in East Asia, migrating from its year-around breeding area in the Indochina Peninsula and South China as far as Northeast China, Japan and the Korean Peninsula [3,7,8,9]. Therefore, it is important to evaluate the flight capability of *S. frugiperda* for accurate monitoring and prediction of the range and timing of its migration, so as to effectively control this migrtory pest and reduce the application of pesticides.

Several studies of the flight ability of *S. frugiperda* have been carried out [10,11,12], but these studies are limited to laboratory-reared populations. These results may reflect the natural migration ability of *S. frugiperda*, but they could also be unrepresentative of the flight ability of populations that developed under natural conditions. Insect populations in the natural environment often face sudden changes of environmental conditions, so the phenotype of wild insects may be quite different from that of indoor populations [13]. Numerous examples have been documented of distinct differences in genetic structure, behavior and biological characteristics between laboratory and field populations of many insects, including flight behavior and capability [14,15,16,17]. For example, field populations of boll weevils (*Anthonomus grandis*) exhibited longer flight distances, flight durations and flight speeds than laboratory-reared strains [18]. In summary, it is more reliable to measure the flight capability of *S. frugiperda* from field populations because it will reflect the species real flight potential to a greater extent than measurements of lab-reared individuals.

Temperature is one of the key environmental factors affecting the flight capability of insects. Both the initiation of migration and persistent migratory flight need appropriate temperature conditions [19]. Insects cannot fly when the air temperature at flight altitude falls below a certain temperature threshold, at which point they must either descend to an altitude with a more suitable temperature or land [20,21]. Thus, low temperature is an important environmental factor causing migrating insects to be concentrated and deposited, a phenomenon known as the low-temperature barrier [22]. Therefore, the low temperature threshold determines the duration and distance of insect migrations, and thus, it is an important component of trajectory simulations for estimating the pathways and endpoints of migratory flights. However, the low temperature threshold varies between different species. For example, the flight temperature threshold of oriental armyworm (*Mythimna separata*) is about 16 °C [21], that of cotton bollworm (*Helicoverpa armigera*) is 13 °C [23], and that of brown planthopper (*Nilaparvata lugens*) is 16.9 °C [20]. The low temperature threshold for the flight of *S. frugiperda* is still unknown. In our previous studies, the flight temperature threshold of *S. frugiperda* was set to 13.8 °C when calculating migration trajectories; however, this is the low temperature threshold for *S. frugiperda* larval development, and it may not be appropriate for determining flight activity [3,24].

In this study, the flight capability of a field population of *S. frugiperda* from South China, and the low temperature threshold for flight was quantified using a tethered-flight mill system. The determination of these parameters from a natural population will help to improve the prediction of the occurrence range and migration timing of *S. frugiperda* in East Asia, information that can be used to inform integrated pest management strategies for this pest.

## 2. Materials and Methods

### 2.1. Determination of the Flight Capability of a Natural Population of S. frugiperda

#### 2.1.1. Collection and Feeding of *S. frugiperda*

In June 2019, mature larvae (5th or 6th instar) or pupae of *S. frugiperda* were collected from a corn field located at the Guangxi Academy of Agricultural Sciences (GAAS) experimental farm (108.06° E, 23.25° N), Wuming District, Nanning City, Guangxi Zhuang Autonomous Region, South China. Each of these larvae were placed in a separate plastic cup, and were placed in an indoor rearing room at the main GAAS facility in Nanning, and fed with corn leaves until they pupated. The rearing room was well ventilated with no extra heating and cooling, and had no equipment running, so that it closely matched the natural ambient temperature conditions of the local environment; it also received ample natural lighting so that the caterpillars and pupae experienced natural photoperiod conditions. Pupae were placed on a layer of dampened cotton to maintain humidity until they emerged, and the cotton was dampened every 3 days. The newly emerged adult was placed into a 250 mL plastic cup with a layer of cotton dipped in 10% honey solution for feeding before testing their flight capability.

#### 2.1.2. Flight Mill and Tethering

In these experiments, we used an 8-channel flight mill system designed by Ka-Sing Lim of Rothamsted Research, UK [25]. This flight mill system is composed of eight separate, circularly-rotating (‘roundabout’) flight mills, capable of testing eight insects at the same time. The data acquisition frequency is every 5 s (i.e., the number of rotations made by the test moth is recorded every 5 s), so that the flight distance and flight speed of insects can be calculated according to the diameter of the flight arm and the number of rotations in the total experimental duration or per unit time. This flight mill system has measured the flight capability of several insect species in previous studies [26,27]. In our experiments, every two flight mills were monitored by an infrared camera to ensure the flight activity of tethered moths was recorded correctly.

Our experiments were carried out during the period from 29 July to 29 September 2019 in a flight chamber in Nanning, which was covered with blackout material so that moths were kept in the dark at all times. All *S. frugiperda* moths were randomly assigned to a flight mill and tethering was started at dusk. The moths were between 1 and 6 days old when the experiment was started; moths were initially kept in a plastic cup and were sedated at 4 °C for a minimum of 2 min prior to tethering. Then, the moth was attached to a pin that fits into a sleeve suspended from the flight arm. The ‘pin’ was a copper wire with a diameter of 0.75 mm and a length of about 2 cm. Before tethering, scales were removed from the thorax and a small amount of adhesive glue was applied to both the ‘pin’ and the exposed thorax. Tethering was completed one hour before dusk, when the experiment was started and moths were then kept in darkness. Recording started when a *S. frugiperda* moth was attached to the flight mill in the tethering chamber and placed into darkness. Prior to attachment, each moth was provided with a small paper ‘platform’ which it grasped in its tarsi, so that it could take off (initiate flight) by dropping the paper when it chose to. The first 56 moths (mainly at 1–3 days old) were tethered until they died (see Table 1). Because most of the moths did not fly after 24 h, the rest of the moths were only tethered for 24 h. The test temperature in the chamber was 25 ± 1 °C and relative humidity was 65% ± 5%, which was typical of local night-time conditions and has previously been determined as the most suitable temperature and humidity for indoor flight activity of *S. frugiperda* [11,12].

#### 2.1.3. Data Analysis

Data were tested to check whether they fitted a normal distribution using the Shapiro–Wilk test before further analysis. The flight speed data conformed to a normal distribution, but neither flight duration nor flight distance did. Therefore, the coefficient of variation of flight duration and distance was calculated, and the flight duration and distance at different ages or genders were compared with Kruskal–Wallis tests and Wilcoxon rank tests for multiple comparisons and pairwise comparisons, respectively. The flight speed of moths at different ages or genders were compared with *t*-tests. All data were analyzed in R (version 4.1.3, https://www.r-project.org/ (accessed on 10 March 2022)).

### 2.2. Determination of the Low Temperature Threshold for Flight of S. frugiperda

#### 2.2.1. Collection and Feeding of *S. frugiperda*

The *S. frugiperda* moths used to determine the low temperature flight threshold in December 2020 were from a lab-reared population in Nanjing, originally collected from a corn field in GAAS in June 2019. This population was maintained on an artificial diet for more than 21 generations. The rearing temperature was 28 ± 1 °C; the relative humidity was 60% ± 5%; the light-dark photoperiod was 14 h–10 h and the light intensity was 18,000 lx.

#### 2.2.2. Flight Mill and Tethering

A 24-channel insect flight mill system (Jiaduoke Industry and Trade Co., Ltd, Hebi City, China) was used to automatically collect the flight data of insects in this experiment in Nanjing. This flight mill system is composed of 24 flight mills for tethering 24 moths at the same time, and its data acquisition frequency is 5 min. This system has been successfully used in the determination of the flight ability of *S. frugiperda* in previous studies [11,12]. The tethering methods were the same as above. In this experiment, only the 2-day-old adults were tested for 12 h flight.

This experiment was carried out in an artificial climate chamber (BD-RSZ-II, Beidi Experimental Instrument Co., Ltd, Nanjing City, China) that was used to simulate a varying temperature environment. Two temperature gradient treatments were set in the artificial climate chamber. Treatment I experienced a gradual cooling, comprised of 6 temperature levels: 20 °C, 18 °C, 16 °C, 14 °C, 12 °C, and 10 °C, with each temperature lasting for 2 h (including a temperature changing time of about 10 min) for a total of 12 h (from 19:00 to 07:00 the next morning). Treatment II experienced a gradual warming, comprised of 6 temperature levels: 10 °C, 12 °C, 14 °C, 16 °C, 18 °C, and 20 °C, with each temperature lasting for 2 h (including the temperature changing time).

#### 2.2.3. Data Analysis

To determine the low temperature threshold at which *S. frugiperda* terminates flight activity, a flight duration ≤5 min during each 2 h was identified as ‘non-flight’ and assigned a value of ‘False’, while a flight duration >5 min was identified as ‘flight’ and assigned a value of ‘True’. Then a logistic generalized linear model was applied in R. The status of flight (True or False, i.e., ‘flight’ or ‘non-flight’) was the response variable, while potential explanatory variables included gender, mean temperature during each 2 h, treatment (decreasing or increasing temperature) and tethering hours since 19:00. A forward selection was applied to identify the last minimum model by using Akaike’s Information Criteria (AIC).

## 3. Results

### 3.1. Flight Capability of a Field Population of S. frugiperda

From 29 July to 29 September 2019, 158 *S. frugiperda* moths (62 females and 96 males) were tethered and flown on the flight mill system (Table 1). 84 of these moths, aged 1–3 days, were left attached to the mill until they died. The majority ceased flight activity after 24 h (median time to termination: 24.55 h; females, 29.89 h; males, 23.31 h), but some individuals still showed flight activity after 48 h (maximum for females, 49.61 h; males, 49.60 h). The median of total flight duration was 6.47 h (5.70 h for females and 7.15 h for males), and the median flight distance was 13,255 m (10,693 m for females and 14,276 m for males). The longest flight duration was 36.51 h for females and 23.34 h for males, and the longest flight distance was 116.729 km for females and 65.354 km for males. These results indicate that some individuals have very high flight capability.

Generally, the proportion of *S. frugiperda* individuals showing flight activity reduced over time, with only 64.6% of moths (102/158) still flying after 5 h. However, there was little change in the next 7 h, as 50.6% of moths (80/158) were still showing flight activity at 12 h, and their flight speed remained relatively stable over this period (Figure 1), indicating that most individuals that show moderate flight activity (at least 5 h) will continue for a complete night’s flight at a constant speed (indicative of migration). Therefore, the data from the first 12 h of the experiment were analyzed further in the following section.

### 3.2. Flight Capability of S. frugiperda Moths in the First 12 h

Over the first 12 h of the experiment, the flight duration varied among individuals. Its frequency distribution was not normally distributed and had a coefficient of variation of 0.66. The mean flight duration was 6.05 ± 0.32 h (mean ± standard error (S.E.), df = 158), and the longest was 12 h. There was a trough in the frequency distribution of flight duration at 5 h, which divided the population into two groups: 58.23% of individuals (92/158) had a flight duration ≥5 h (Figure 2A). Kruskal–Wallis tests showed that there was no significant difference among individuals at different ages (χ^2^ = 9.36, df = 5, *p* = 0.096) or genders (χ^2^ = 0.73, df = 1, *p* = 0.392). However, the frequency distribution of flight duration of 1–3-day-old adults was more dispersed, with a greater coefficient of variation (0.81) than that of 4–6-day-old adults (0.51); the mean flight duration of the former was also significantly lower than that of the latter (1–3-day-old adults: 5.15 ± 0.45 h; 4–6-day-old adults: 7.08 ± 0.42 h; Wilcoxon rank sum test: w = 3910, *p* = 0.005).

Flight distance also varied greatly between individuals, and its frequency distribution was skewed with a long tail and a large coefficient of variation (0.82). The mean flight distance of all individuals was 16.09 ± 1.05 km (mean ± S.E., df = 158), and the longest flight was 59.02 km (Figure 2B). Kruskal–Wallis tests showed that there were significant differences in flight distance among individuals at different ages (χ^2^ = 11.61, df = 5, *p* = 0.040) but not between females and males (χ^2^ = 1.74, df = 1, *p* = 0.187). In detail, the mean sustained flight distance of 4–6-day-olds was 25.49 ± 1.51 km, which was significantly longer than that of 1–3-day-old adults (18.54 ± 1.63 km; Wilcoxon rank test: w = 4027, *p* = 0.001). The flight distance of the individuals whose flight duration was longer than 5 h (32.86 ± 0.69 km) was significantly larger than that of the individuals whose flight duration was less than 5 h (6.36 ± 0.64 km) (Wilcoxon rank sum test: w = 68, *p* < 0.001) (Figure 2D).

Flight speed was normally distributed, with a mean speed of 0.68 ± 0.02 m/s (mean ± S.E., df = 158) (Figure 2C). There was no significant difference in flight speed among individuals at different ages (F = 0.79, df = 5, *p* = 0.560) or genders (F = 2.8, df = 1, *p* = 0.096). The flight speed of individuals with flight duration ≥5 h (0.75 ± 0.03 m/s) was much higher than that of individuals with flight duration <5 h (0.58 ± 0.04 m/s; *t*-test: *t* = 3.78, *p* < 0.001; Figure 2E). Moreover, the maximum flight speed of individuals with flight duration ≥5 h was 1.64 ± 0.05 m/s, much higher than that of individuals with flight duration <5 h (1.25 ± 0.06 m/s) (*t*-test: *t* = 5.02, *p* < 0.001; Figure 2F).

### 3.3. Flight Capability of S. frugiperda Moths under Low Temperature

In total, 109 moths (55 females and 54 males) were tethered and tested by flight mill under low temperature condition of 10–20 °C. Among these, 68 moths were in Treatment I, in which the temperature was increased by 2 °C every two hours from 10 °C to 20 °C. The other 41 moths were in Treatment II, in which the temperature declined from 20–10 °C by 2 °C every two hours (Figure 3A,B).

Only 36.7% of moths (40/109) showed any flight activity, with all other individuals staying still during the experiments. Thus, in these comparatively low temperature conditions, the majority of moths were inhibited to fly. As temperatures reduced, more moths stopped flying, and their flight duration and flight speed also decreased (Figure 3A,B). In Treatment I (increasing temperature treatment), flight speed was <0.03 m/s and flight duration was <25 min in all 2-h experimental periods when temperatures were ≤16 °C, and no individuals flew when the temperature was between 10 and 12 °C (Figure 3A,B). However, three moths in Treatment II (declining temperature treatment) were still flying around 10 °C with a speed of 0.04 m/s (Figure 3A,B). Among these three moths, one moth kept flying for 40 min with a speed of 0.08 m/s. Thus, moths experiencing the declining temperature condition had stronger flight capability or stronger flying tendency than moths experiencing increasing temperature.

To identify the low temperature threshold of flight of *S. frugiperda*, a logit regression model was built. The variables of temperature (coefficient = 0.419, z = 7.16, *p* < 0.0001) and ‘treatment’ (coefficient = −0.779, z = 2.89, *p* = 0.0038) affected significantly the probability of moth flight. Here, ‘treatment’ is a logical variable of two values (TRUE and FALSE), where TRUE means an ‘increasing temperature’ treatment. Gender and tethering hour were excluded in the final model, indicating these two variables did not have a significant effect on whether moths fly. Based on the last minimum model, 14.9 °C and 13.1 °C were obtained as the threshold temperature of flight of *S. frugiperda* under increasing and declining temperature conditions, respectively, if we set 0.05 as a threshold value of the probability of whether a moth will fly or not.

## 4. Discussion

*Spodoptera frugiperda* clearly has a strong potential flight ability. In our study, the mean flight speed of the moth is similar to that recorded in previous tethered-flight studies [11,12,27,28], slightly less than 3 km/h (~0.83 m/s). These values are considerably lower than is expected of a free-flying noctuid moth of this size, as similar moths (e.g., oriental armyworm and cotton bollworm) have self-powered airspeeds of about 2.5–3.5 m/s; the difference is likely due to the mechanical friction in our flight-mill system causing tethered moths to fly considerably slower than they would be expected to in nature [27]. A previous study of *S. frugiperda* flight potential indicated that tethered moths can fly for up to 20.56 h, traversing a self-powered distance of 62.98 km [11]. Our study has expanded these limits, revealing that *S. frugiperda* can fly for up to 36.5 h within a 48-h period, covering a distance of 116.7 km. Given the fact that self-powered airspeeds are significantly depressed due to the tethering system, the distances covered during free flight of similar duration would be much greater (of the order of 300–400 km) even without wind assistance. As *S. frugiperda* habitually migrates at high altitude, where fast airflows will facilitate rapid transport [3,28], it is clear that this species will likely migrate several hundred kilometers during its migratory flights. In addition, we found that the flight ability of females is higher than that of males in *S. frugiperda*, and this phenomenon is quite common in other insects, such as *C*. *medinalis*, *M. separata* and *Pseudaletia unipuncta* [26,29,30,31]. This may be because the female moth stores more energy substances, such as fat, which enable the moth to maintain a strong flight ability without feeding.

Our results clearly show there is large individual variation in the flight capability of the South China field population of *S. frugiperda* we investigated. Flight duration of individuals could be clearly divided into two groups, ‘<5 h’ and ‘≥5 h’, with the latter group containing individuals showing a strong migratory tendency. Although several previous studies have studied the flight ability of this species [11,12,27], our study is the first to show that field populations have a clear differentiation between two categories, that we classify as (i) strong migrants, and (ii) resident or weak migrants. The pest moth *Cnaphalocrocis medinalis* can similarly be divided into three categories (resident type, migratory type and strong migratory type) according to the flight duration of individuals quantified by tethered flight [29]. However, it is possible that these categories are not genetically fixed in *S. frugiperda* at the point of pupal emergence, as in the oriental armyworm (*M. separata*), optimal environmental factors (such as sufficient food, and suitable temperature, humidity and light conditions for breeding) can promote most individuals to transform from potential migrants into residents [32].

Flight low temperature threshold is one of the important parameters of insect migration. Although *S. frugiperda* has a wide range of suitable flight temperatures [2,12], the low temperature threshold for flight of *S. frugiperda* is not clear. In this study, it was found that the low temperature threshold of flight was 13.1 °C and 14.9 °C under treatments of decreasing and increasing temperatures, respectively. Here, the low temperature threshold under decreasing temperature was lower. Presumably, this is because flying moths have a stronger willingness to escape the cooling conditions and their body might be warmer than the surrounding temperature when they have flight activity. Compared with other insect pests that have been well studied, the flight low temperature threshold of *S. frugiperda* was lower than that of *N. lugens* and *M. separata* [20,21], and similar to that of *H. armigera* [23]. Moreover, radar studies have shown that the altitude at which nocturnal migrants form layers is sometimes at the height of the low temperature threshold, that is, a ‘ceiling’ layer [20,21,23]. This may mean that layers of migrating *S. frugiperda* may form at similar altitudes to that of migrant *H. armigera*. Previously, the developmental threshold temperature of 13.8 °C was adopted as the flight low temperature threshold in the trajectory simulation in our previous studies [3,7,8,9,33]. As 13.8 °C is rather similar to the flight low temperature threshold of 13.1 °C quantified in this study under a declining temperature treatment, the migration distance and the distribution of *S. frugiperda* predicted in previous studies can be considered reliable [3,7,8,9,33].

## 5. Conclusions

In conclusion, our results demonstrate that the potential migration duration (and thus distance) of a field population of *S. frugiperda* from South China is significantly greater than previously estimated. Furthermore, our study documented the responses of *S. frugiperda* flight behavior to temperature changes, and the flight low temperature threshold of *S. frugiperda* was defined. The results of this paper will be helpful to further understand the migratory rules and flight behavior of *S. frugiperda*, and to provide a theoretical basis for pest forecasting and pest control.

## Figures and Tables

**Figure 1 insects-13-00422-f001:**
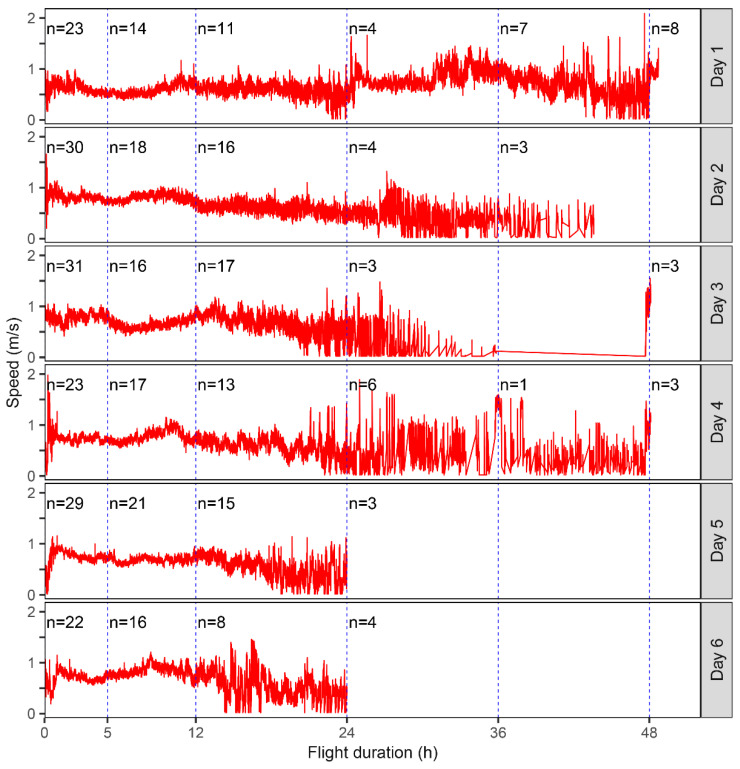
Flight speed and flight duration of *S. frugiperda* of different ages.

**Figure 2 insects-13-00422-f002:**
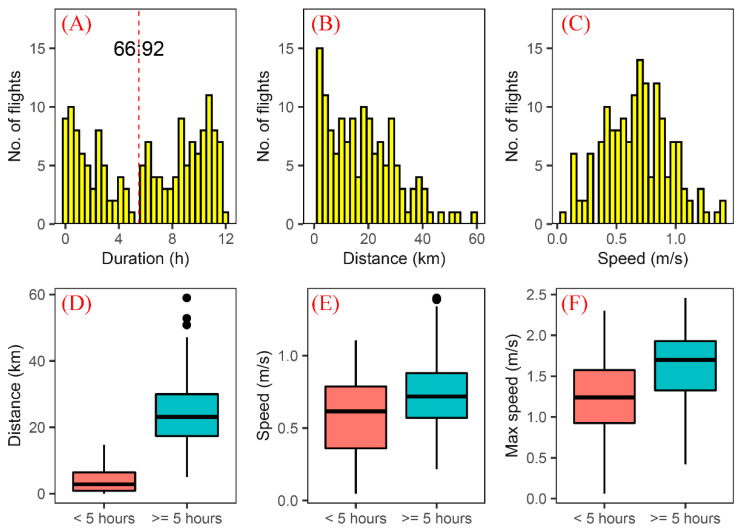
Flight capability of *S. frugiperda* moths in the first 12 h of testing. The distribution of (**A**) flight duration, (**B**) distance and (**C**) speed. Based on the flight duration, the experimental moths can be divided into two groups, one with flight duration of <5 h, and another with flight duration ≥5 h. Compared with the former group, the latter had stronger flight capability, a higher mean flight speed (**E**) and maximum flight speed (**F**), and thus, they covered longer flight distances (**D**).

**Figure 3 insects-13-00422-f003:**
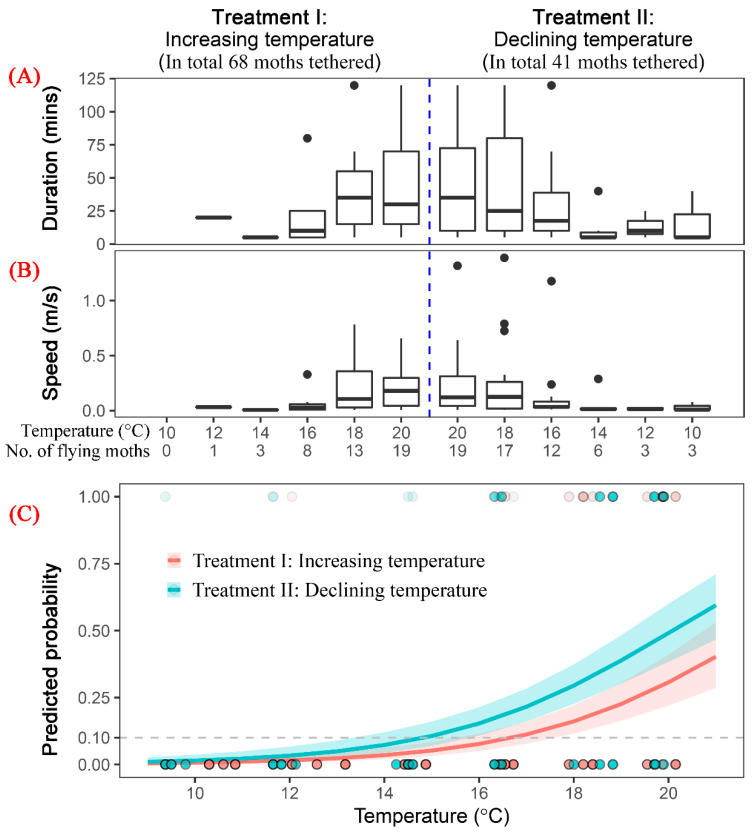
The flight duration (**A**) and speed (**B**) of *S. frugiperda* under different temperatures. (**C**) The smooth curve of logistic GLM models.

**Table 1 insects-13-00422-t001:** Summary of tethered *S. frugiperda* at different ages.

Age	No. of Moths (Total/Female)	Tethering Mode	Max/Min Test Duration (h)	Max/Min Flight Duration (h)
1	23/13	Tethered until death	49.61/26.32	36.51/8.86
2	30/16	Tethered until death	43.60/24.52	33.77/7.32
3	31/10	20 adults tethered until death (7 females); remainder tethered for 24 h	49.00/23.26	22.29/6.19
4	23/7	3 adults tethered until death (2 females); remainder tethered for 24 h	24.00(49.00)/23.69	21.76(22.00)/9.45
5	29/12	Tethered for 24 h	24.00/23.00	19.60/9.31
6	22/4	Tethered for 24 h (5 died before 24 h)	24.00/18.09	16.46/10.11

## Data Availability

The data presented in this study are available from the corresponding author upon request.

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
