# Peer review of "Flight Capability and the Low Temperature Threshold of a Chinese Field Population of the Fall Armyworm Spodoptera frugiperda"

_insects, 2022, doi:10.3390/insects13050422_

Round 1
Reviewer 1 Report
This is great work and very helpful in understanding S. frugiperda flight and migration. There are no necessary revisions from my standpoint. The manuscript very clearly and succinctly introduces the fundamental knowledge about S. frugiperda flight and the use of flight mills. Then gives an appropriate description of the methods. The results are succinct and the discussion does not stray from the results. It is a very good foundational work that can be used to build further knowledge about S. frugiperda flight.
Author Response
Thanks very much.

Reviewer 2 Report
Carlos A. Blanco wrote:
This is a good contribution to our knowledge of the fall armyworm. It is well thought out and I hardly found any mistakes or significant ‘voids’.
One aspect that may help the reader is that in numerous occasions has been repeated that “the finding of this research would be useful for control tactics -or similar language-“. I agree with that, but I may be imagining something different to what the authors have in mind. Explain why this is important.
Of really minor issues:
Line 22: 58% of iIndividuals (58%) have a strong…
Lines 55-56: Provide reference(s) for Spodoptera frugiperda cannot survive winter periods in temperate and subtropical regions.
Lines 72-74. I really do not see the relevance of mentioning the boll weevil as a good example.
Lines 80-82: Provide reference(s) for Insects cannot fly when the air temperature at flight altitude falls below their low temperature threshold at which flight is possible, at which point they must either descend to an altitude with a more suitable temperature or land.
Author Response
Comment 1: One aspect that may help the reader is that in numerous occasions has been repeated that “the finding of this research would be useful for control tactics -or similar language-“. I agree with that, but I may be imagining something different to what the authors have in mind. Explain why this is important.
Answer 1: Thanks. Migratory insects have certain autonomous flight ability in the air. At present, the prediction of the immigration and emigration of the S. frugiperda needs to be combined with the meteorological conditions and its own flight behavior. Therefore, under the same meteorological conditions, the more accurate quantification of its own flight behavior can more effectively improve the accuracy of prediction and explore its migration pattern. For example, when the trajectory model is used to predict its trajectory, the flight speed of the S. frugiperda itself will affect the effective landing site of the trajectory, and finally affect the simulation of the landing area of the population (a more accurate prediction of the landing area). The flight low temperature threshold can also narrow the altitude simulation range and make the trajectory simulation altitude more accurate. The flight duration can provide us with some physiological basis for the migration of insects across the sea or in some extreme cases. All of which are very helpful to predict the migration S. frugiperda and the accurate predictions of a great method to deal with the sudden outbreak characteristics of migratory pests.
Comment 2: Line 22: "58% of individuals (58%) have a strong…" change to "Individuals (58%) have a strong…"
Answer 2: Changed.
Comment 3: Lines 55-56: Provide reference(s) for Spodoptera frugiperda cannot survive winter periods in temperate and subtropical regions.
Answer 3: Added
Comment 4: Lines 72-74. I really do not see the relevance of mentioning the boll weevil as a good example.
Answer 4: Thank you! Here I try to say that the field population has a stronger flight ability than the laboratory population among insects.
Comment 4: Lines 80-82. Provide reference(s) for Insects cannot fly when the air temperature at flight altitude falls below their low temperature threshold at which flight is possible, at which point they must either descend to an altitude with a more suitable temperature or land.
Answer: added.

Reviewer 3 Report
Reviewer comments
This work experimentally investigates the flight capability and low temperature threshold of the fall armyworms (Spodoptera frugiperda), a Chinese field population. The results indicated that the flight capability of S. frugiperda moths can vary greatly between individuals, and that some adults are capable of flying great distances. Furthermore, the low temperature flight threshold of the moths is 13.1C under declining temperatures. I think this work, to some extent, is of interest and shows some certain scientific values. It can be given a major revision and can be accepted for publication in the journal if authors can address the following comments.
Comments:
Line 199: How did the flight mill system measure the flight distance of the moths? These no evidence for relevant description in Material and Methods.
Line 196-203: Could you give a potential explanation for the flight capability variation between the female and male moths that the longest flight duration and flight distance of the females are higher than that of the males?
Line 299: “km” should be kilometers.
Line 317: Why the low temperature threshold 13.1℃ at decreasing temperatures is lower than that (14.9℃) at increasing temperatures? Please give a potential explanation.
Author Response
Comments 1: Line 199: How did the flight mill system measure the flight distance of the moths? These no evidence for relevant description in Material and Methods.
Answer 1: (Line 124-126) “the flight distance and flight speed of insects can be calculated according to the diameter of the flight arm and the number of rotations in total experimental duration or in per unit time.”
Comments 2: Line 196-203: Could you give a potential explanation for the flight capability variation between the female and male moths that the longest flight duration and flight distance of the females are higher than that of the males?
Answer 2: Thank you! Here is my explanation trying to explain it: We didn't supplement energy materials during the flight. So the reason why the female flight ability is stronger may be that the female moth stores more energy substances such as fat, which can still maintain a strong flight ability without feeding.
Lines 305-309: ” In addition, we found that the flight ability of females is higher than that of males in S. frugiperda, and this phenomenon is quite common in other insects, such as C. medinalis, M. separata and Pseudaletia unipuncta [26,29-31]. This may be because the female moth stores more energy substances, such as fat, which can still maintain a strong flight ability without feeding.”
Comments 3: Line 299: “km” should be kilometers.
Answer 3: Changed.
Comments4: Line 317: Why the low temperature threshold 13.1℃ at decreasing temperatures is lower than that (14.9℃) at increasing temperatures? Please give a potential explanation.
Answer4: (Line 328-331) ‘Here, the low temperature threshold under decreasing temperature was lower, this presumably is because flying moths have a stronger willingness to escape the cooling condition, and their body might be warmer than the surrounding temperature when they have fly activity.’

Round 2
Reviewer 3 Report
The manuscript can be accepted for publication in Insects.